# Non-communicable diseases (NCDs) and vulnerability to COVID-19: The case of adult patients with hypertension or diabetes mellitus in Gamo, Gofa, and South Omo zones in Southern Ethiopia

**Fikre Bojola[1], Wondimagegn Taye[2], Habtamu Samuel[2], Bahiru Mulatu[2], Aknaw Kawza[3], Aleme Mekuria[2]***

**1** Department of Clinical nursing, Arba Minch College of Health Sciences, Arba Minch, Ethiopia,
**2** Department of public health, Arba Minch College of Health Sciences, Arba Minch, Ethiopia, **3** Southern Region Health Office, Ethiopia

* alemmekurishet@gmail.com

## Abstract

### Background

A growing body of evidence demonstrating that individuals with Non-Communicable Disease (NCD) are more likely to have severe forms of COVID-19 and subsequent mortality. Hence, our study aimed to assess the knowledge of vulnerability and preventive practices towards COVID-19 among patients with hypertension or diabetes in Southern Ethiopia.

### Objective

To assess the knowledge and preventive practices towards COVID-19 among patients with hypertension or diabetes mellitus in three zones of Southern Ethiopia, 2020.

### Methods

A community-based cross-sectional study design was used with a multi-stage random sampling technique to select 682 patients with hypertension or diabetes mellitus from 10th -17th July 2020 at the three zones of Southern Ethiopia. Logistic regression analysis with a 95% confidence interval was fitted to identify independent predictors of knowledge and preventive practices towards COVID-19. The adjusted odds ratio (AOR) was used to determine the magnitude of the association between the outcome and independent variables. P-value <0.05 is considered statistically significant.

### Results

The Multi-dimensional knowledge (MDK) analysis of COVID-19 revealed that 63% of study subjects had good knowledge about COVID-19. The overall preventive practice towards COVID -19 was 26.4%. Monthly income (AOR = 1.42; 95% CI: 1.04, 1.94) significantly

**Data Availability Statement:** All relevant data are within the paper and its Supporting Information files.

**Funding:** The authors received no specific funding for this work.

**Competing interests:** The authors have declared that no competing interests exist.

predicted knowledge towards COVID-19. Ninety-five percent of the study subjects knew that the COVID-19 virus spreads via respiratory droplets of infected individuals. One hundred and ten (16.2%) of study subjects correctly responded to the questions that state whether people with the COVID-19 virus who do not have a fever can infect the other. Knowledge about COVID-19 (AOR = 1.47; 95% CI: 1.03, 2.1) became the independent predictor of preventive practice.

## Conclusions

In this study, the knowledge of the respondents towards the COVID-19 pandemic was good. But the preventive practice was very low. There was a significant gap between knowledge and preventive practices towards the COVID-19 pandemic among the study subjects. Monthly income was significantly associated with knowledge of COVID-19. Knowledge of COVID-19 was found to be an independent predictor of preventive practice towards COVID-19. Community mobilization and improving COVID-19- related knowledge and practice are urgently recommended for those patients with hypertension or diabetes mellitus.

## Introduction

Non-communicable diseases (NCDs) are becoming a major health care challenge in the world. They are presently competing with traditionally leading killer diseases in the death toll. The topmost killer NCDs are; various types of cancers, chronic respiratory illnesses, stroke, and cardiovascular diseases [1]. Globally, ischemic heart disease, stroke, and Chronically Obstructive Pulmonary Diseases (COPD) are the three leading causes of mortality. Non-communicable diseases (NCDs) constitute close to 54% of the overall disease burden, as measured in disability-adjusted life years (DALYs) [2].

Non-communicable diseases and associated risk factors (unhealthy diets, physical inactivity, harmful use of alcohol) are on the rise in developing countries, posing a threat to the health and financial systems of emerging economies [2]. Because non-communicable diseases have not been received equal attention with communicable diseases in middle-income and low-income countries, people of these countries are disproportionately suffering from the consequences of these diseases [3, 4].

According to the World Health Organization (WHO) 2014 country profile, about 30% of total deaths in Ethiopia were associated with NCDs from which cardiovascular diseases, cancers, chronic respiratory diseases, and diabetes are the leading causes of morbidity and mortality. Similarly, the report revealed disproportionate age-specific death rates with a significant rise in deaths from non-communicable diseases between the ages 30 and 70 years [5].

Globally, the overall Case Fatality Rates of COVID-19 vary between countries. For instance, 4.1% in China, 4.6% in Spain, 8.3% in Italy,2.73% in Egypt, and 1.6% in Ethiopia [6]. The fatality rate of patients with COVID-19 was highest in in-person aged 80, ranging from 13% to 16.7%, followed by 7.2%–8.9% among those aged 70–79 years [7]. However, patients with NCDs are more likely to have severe disease and subsequent mortality. The COVID-19 pandemic has had widespread health impacts, revealing the particular vulnerability of those with underlying conditions [8].

The most commonly reported non-communicable diseases that complicate COVID-19 and lead to increased morbidity and mortality are: diabetes mellitus (DM), cardiovascular diseases

like hypertension, cerebrovascular disease, coronary artery disease (CAD), and respiratory disease like chronic obstructive pulmonary disease (COPD) and tuberculosis [9].

Physical distancing or quarantine can lead to poor management of NCD behavioral risk factors, including unhealthy diet, physical inactivity, tobacco. Patients living with NCDs are at increased risk of the health impacts of emergencies such as COVID-19 [10].

An association between COVID-19 severity and NCDs has also been reported in China and the USA. However, many COVID-19 deaths also occur in older people who often have existing comorbidities [11]. Body-mass index (BMI) might also be associated with the severity of COVID-19; in China, patients with severe COVID-19 and non-survivors typically had a high BMI (>25 kg/m$^2$). The impact of COVID-19 response measures on NCDs is multifaceted [12].

A study conducted on COVID-19 in Wuhan, China, showed that from 52 intensive care unit patients with novel coronavirus disease, 22% had cerebrovascular diseases and 22% had diabetes. The Same study in Wuhan revealed that out of 1099 patients with confirmed COVID-19, of whom the quarter had hypertension, 16·2% had diabetes mellitus [13]. In another study in the same place in China, out of 140 patients admitted to the hospital with COVID-19, 30% had hypertension, and 12% had diabetes [9].

Patients with NCDs are a highly affected group during the ongoing COVID -19 epidemic due to loss of jobs and wages coupled with disruptions in their usual sources of drug access. Moreover, non-adherent patients having NCDs have a manifold higher risk of complications resulting from uncontrolled disease [14]. NCD patients may, therefore, continue to be at persistent risk of COVID -19 while attending Primary Health Care (PHC)/Comprehensive Health Care (CHC) for meeting their health requirements. The failure to address and sufficiently resolve the barriers in attaining acceptable levels of care and management of patients having NCDs at the time of the COVID -19 pandemic represents a grave public health concern [15].

The survey report of WHO indicates that, due to the COVID -19 pandemic among 155 countries, 120 countries reported that NCD services are disrupted. The main factors related to service disruption are: a decrease in patient volume due to cancellation of elective care, closure of population-level screening programs, government or public transport lock-downs hindering access to the health facilities, NCD related clinical staff deployed to provide COVID-19 relief, and closure of outpatient NCD services as per government directive. This disruption of routine health services and medical supplies risks increasing morbidity, disability, and avoidable mortality over time in NCD patients [16].

Studies conducted regarding COVID-19 with NCDs in Ethiopia are very limited. Therefore, this study aimed to assess the knowledge and preventive practices towards COVID-19 among people with hypertension and diabetes mellitus in the Gamo, Gofa, and South Omo zones of Southern Ethiopia.

## Methods and materials

### Study setting

The study was conducted in three Zones of Southern Ethiopia namely; Gamo, Gofa, and South Omo Zone. Arba Minch, the capital town of Gamo zone, is located 505 km south of Addis Ababa, the capital city of Ethiopia, and 275 km Southwest of Hawassa, capital of the South region. The total population of the town based on the 2007 census is 112,724. Of which the total number of women comprises 56,908. Sawla is a town of the newly established Gofa Zone, which is 500 km south of Addis Ababa and it has a population of 60,000. South Omo is a Zone of pastoralists in the region. Jinka is the capital with a total population of 37,000 [17]. [**Fig 1**].

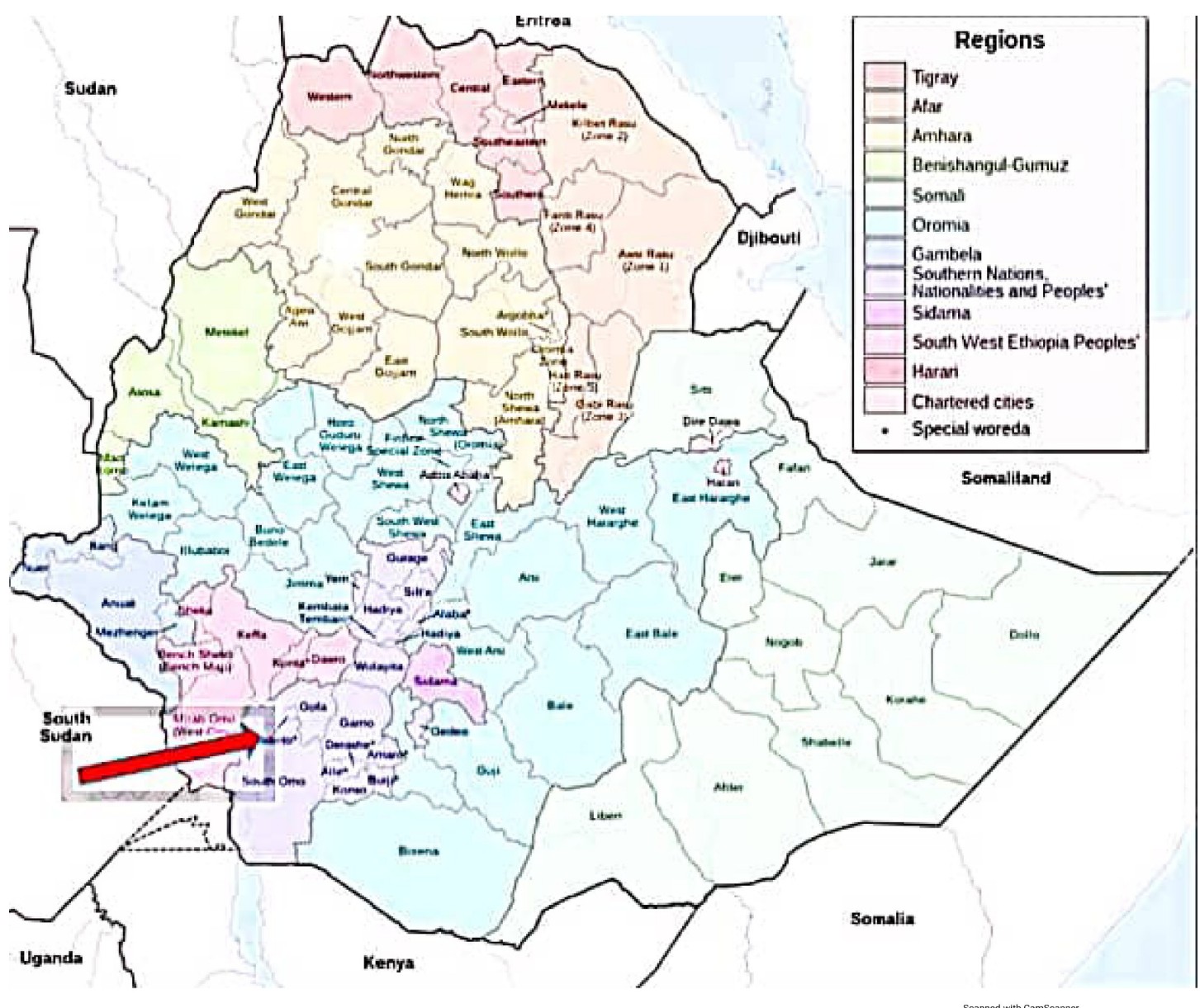

**Fig 1. Map of the study area (Gamo, Gofa, and South Omo zones), Southern Ethiopia.**

## Study design and period

A community-based cross-sectional study was conducted from 10th -17th July 2020 to assess the level of knowledge and preventive practice towards COVID-19 and assess predictors among patients with hypertension or diabetes mellitus.

## Population and sample

Before sampling, chronic patients' follow-up data records of each of the three zonal hospitals (Arba Minch, Sawla, and Jinka) were reviewed thoroughly for different types of non-communicable diseases. However, the number of other NCD cases was insignificant to sampling. Therefore, we made the sampling frame only for patients with hypertension or diabetes

mellitus. Hence, this study was conducted among randomly selected people with diabetes mellitus or hypertension patients. Patients who had a serious illness and who cannot communicate during data collection were excluded from the study.

## Sample size & sampling procedure

We calculated the sample size for this study using a single population proportion formula by using a proportion of knowledgeable visitors about COVID-19 (72%) from a study conducted in Jimma University Medical College, Ethiopia [18]. Ninety-five percent certainty and 5% margin of error between populations and samples with a non-response rate of 5% and design effect of two.

Therefore, the total calculated sample size was 704 hypertensive or diabetic patients. In Gamo Zone, there were four town administrations, among which Arba Minch town was selected randomly using the lottery method. In Gofa Zone, there were two town administrations and Sawla town was randomly selected. From the South Omo Zone, Jinka town was selected in the same manner.

Data about the study participants were collected by reviewing hospitals' follow-up clinic databases. A list of patients who were diagnosed with diabetes mellitus or hypertension was included in the sampling frame. Then, a simple random sampling technique was used to select the study participants until the calculated sample size was achieved. Finally, using the names of the selected patients, their addresses were sought, traced, and interviewed at the home level [**Fig 2**].

## Instrument and measurement

The knowledge and preventive practices towards the COVID-19 were measured using tools adapted from WHO and other resources [19, 20]. This study used a descriptive statistic to summarize the knowledge and preventive practices of hypertensive or diabetic patients towards the coronavirus pandemic. The data were collected using a pre-tested, structured interviewer-administered questionnaire. The questionnaire included socio-demographic characteristics, knowledge, and preventive practices towards COVID-19. The questions assessing knowledge (14 questions) were answered on a correct/not correct basis. A correct answer was assigned 1 point and an incorrect/unknown answer was assigned 0 points. The total knowledge score ranged from 0 to 14. Participants' overall knowledge was categorized. Therefore, we used the cut-off point as 'poor' if the score was less than 60% ($< 8$ of 14 points) and 'good' if the score was above 60% ($> 8$ of 14 points). Similarly, the questions assessing practice (9 questions) were answered 'yes' or 'no', the correct answer was assigned 1 point and an incorrect answer was assigned 0 points. The overall preventive practice score was categorized using the same parameter. If the overall score for preventive practice is less than 60% ($<7$ of 9 points) is 'poor' and 'good' if the score is more than 60% ($> 7$ of 9 points). The preventive practice component consists of questions about: frequently washed hands with water and soap, stopped shaking hands while giving a greeting, avoided proximity including while greeting (within 2 meters), have not been gone to a crowded place, using a face mask when leaving home, avoided touching eye, nose, mouth before washing hands, used cover/elbow during coughing/sneezing, used sanitizer (alcohol-rubbing, no contact with surfaces), and have stayed at home.

## Data collection, management, and analysis

The data were collected using a structured, standardized, pre-tested interviewer-administered questionnaire. The questionnaire was developed in English and then translated into Amharic (the local language) then back-translated to the English language for its consistency by two

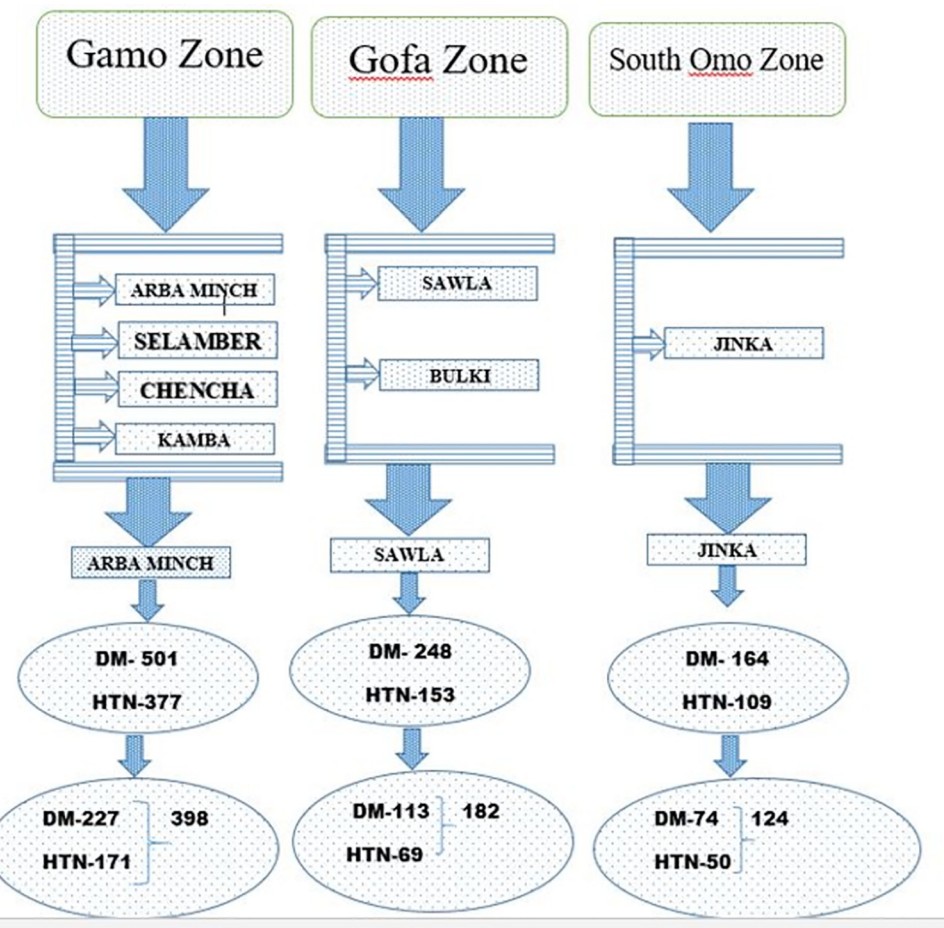

**Fig 2. Schematic presentation of sampling procedure.**

different language experts who speak both English and Amharic fluently. Pre-testing of the questionnaire was done on 5% of the calculated sample among hypertensive or diabetic patients who were not being included in the study. To ensure reliable data collection and attain standardization, the reliability of the knowledge and practice questions were checked. Cronbach's alpha was computed and its value was 0.76 and 0.83 for knowledge and preventive practices, respectively.

Twelve nurses (who had experience in data collection) were recruited and given training on data collection procedures. In addition, twelve home guides, who know the participants' homes in each kebele (village), were used along with data collectors. Three Public Health experts were employed to supervise the daily data collection process.

During the data collection process, data collectors and home guides were wearing medical masks and used sanitizer to rub their hands. To be safe for both the data collector and the respondent, a distance of 2 meters in between was kept. Data were entered into Epi Info version 7.00 software and then exported to SPSS version 25 statistical package for analysis. Descriptive statistics were done and summarized by the frequencies and proportions for categorical predictors.

The outcome variables were dichotomized as 0 = no and 1 = yes. Bivariate analysis was carried out to see the crude effect of each independent variable on the outcome variable. Associations with a p-value <0.05 were considered statistically significant. These variables with a p-

value < 0.25 in the bivariate analysis were candidates for multivariate logistic regression analysis.

To control confounding effects and identify the independent predictors, multivariate logistic regression analysis was done using a backward stepwise variable selection method (backward LR). Hosmer and Lemeshow's goodness of fitness test was used to check for model fitness. Multicollinearity among independent variables was checked using the Variance Inflation Factor (VIF>10). Adjusted odds ratio with its 95% confidence interval was used to identify factors independently associated with the outcome variable and p-values < 0.05 were considered for statistical significance.

### Ethical approval and considerations

Ethical approval was obtained from Arba Minch College of Health Sciences Institutional Review Board (IRB). Support letters were obtained from the three zonal health departments (Gamo, South Omo, and Gofa zones), town administrations, and health offices. The purpose of the study was explained to the study participants. Oral and written consent was secured before data collection. Confidentiality of the information was also ensured.

### Operational definition

**Poverty line.** Is a level of personal or family income below which one is classified as poor according to governmental standards. The cut-off point for income rate is categorized based on the indicator of the proportion of the population below the international poverty line, which is defined as the percentage of the population living on less than $1.90 a day at international prices. The 'international poverty line ' is currently set at $1.90 a day at international prices. We computed the income variable by multiplying $1.90 by the current exchange rate (Ethiopian Birr currency). Therefore, the cut-off point is 2500 ETB per month; meaning <2500 ETB (below poverty line) and ≥ 2500 ETB (above poverty line).

## Results

### Socio-demographic characteristics of the respondents

A total of 678 respondents willingly participated in this study, yielding a response rate of 96.3%. The majority of the respondents, 256 (37.8%) and 282 (41.6%) were within the age range of 31–50 and 51–64 years, respectively. The mean age was 54 ± 11.25 SD years. Among the total respondents, 436 (64.3%) were males and 513 (75.7%) were married. Orthodox Christianity was the most frequent religion 463 (68.3%), followed by protestants 141 (20.8%). Three hundred and twenty-four respondents (47.8%) had a monthly family income of less than 2500 ETB with a median income of 2,900 ETB. Concerning educational status, two hundred and twenty-one (32.6%) achieved secondary school and above. Two hundred and thirteen (31.4%) respondents were government employees, and 48 (7.1%) were housewives. Three hundred and ninety respondents (57.5%) were from Arba Minch town [**Table 1**].

### Knowledge of COVID-19 among patients with hypertension or diabetes mellitus

The Multidimensional knowledge (MDK) analysis of COVID-19 revealed that 427(63%) of respondents had good knowledge of COVID-19 [**Fig 3**].

From the specific MDK questions, 412 (60.8%) of the respondents reported correctly that the main clinical symptoms of COVID-19 are fever, fatigue, dry cough, and myalgia. Regarding the knowledge of high risk about COVID-19, 468 (69%) patients correctly responded

**Table 1. Socio-demographic characteristics of respondents, Southern Ethiopia, July 2020.**

| Variables | Category | Frequency(n) | Percent (%) |
|---|---|---|---|
| Age | 18–30 | 11 | 1.6 |
| | 31–50 | 256 | 37.8 |
| | 51–64 | 282 | 41.6 |
| | 65–80 | 127 | 18.7 |
| | 80+ | 2 | 0.3 |
| Sex | Male | 436 | 64.3 |
| | Female | 242 | 35.7 |
| Religion | Orthodox | 463 | 68.3 |
| | Protestant | 141 | 20.8 |
| | Muslim | 60 | 8.8 |
| | Others | 14 | 2.1 |
| Educational level | Cannot read and write | 101 | 14.9 |
| | Can read and write | 63 | 9.3 |
| | Primary school complete | 90 | 13.3 |
| | Secondary school complete | 203 | 29.9 |
| | Certificate and above | 221 | 32.6 |
| Residence | Arba Minch Town | 390 | 57.5 |
| | Sawla Town | 171 | 25.2 |
| | Jinka Town | 117 | 17.3 |
| Occupation | Government Employee | 213 | 31.4 |
| | Pensioner | 119 | 17.6 |
| | Merchant | 118 | 17.4 |
| | NGO/Private Employee | 82 | 12.1 |
| | Housewife | 48 | 7.1 |
| | Others | 98 | 14.5 |
| Income | < 2500 | 324 | 47.8 |
| | ≥ 2500 | 354 | 52.2 |

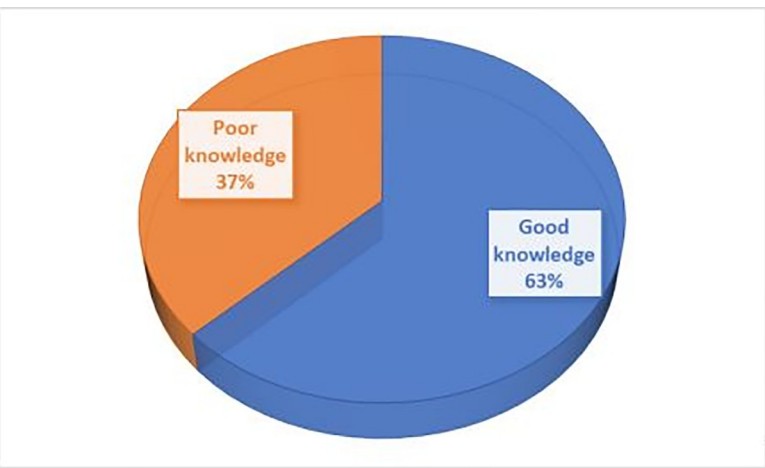

**Fig 3. Multidimensional knowledge (MDK) status of COVID-19 among patients with hypertension or diabetes mellitus, Southern Ethiopia, July 2020.**

whether all persons with COVID-19 will develop severe cases and those who are elderly, have chronic illnesses & are obese are more likely to develop severe cases. Ninety-five percent of the study subjects knew that the COVID-19 virus spreads via respiratory droplets of infected individuals. For questions assessing knowledge about the mode of transmission and infectiousness, only 110 (16.2%) of patients correctly responded to the question that states whether people with the COVID-19 virus who do not have a fever can infect the other. From the components of knowledge about the ways of prevention, 640 (94.4%) responded that individuals should not go to crowded places or avoid taking public transports [Table 2].

## The respondents' preventive practices towards COVID-19

The majority, 499 (73.6%) respondents had poor practice towards prevention of COVID-19 [Fig 4]. Concerning the specific preventive practices, 379 (54.9%) were not frequently washing their hands,240 (35.4%) were not using sanitizer or alcohol rubs. Two hundred and forty-five (36.1%) respondents were still not using face masks [Table 3].

## Awareness of vulnerability towards COVID-19 among patients with hypertension or diabetes mellitus

Among the total respondents, 666 (98.2%) heard about the COVID-19 pandemic. Concerning vulnerability, 595 (87.8%) knew that people with non-communicable diseases are vulnerable

**Table 2. Knowledge of COVID-19 among people with hypertension or diabetes mellitus, Southern Ethiopia, July 2020.**

| Knowledge variables | Frequency | | | |
|---|---|---|---|---|
| | Correct | | Not correct | |
| **Knowledge of symptoms** | No. | % | No | % |
| The main clinical symptoms of COVID-19 are fever, fatigue, dry cough, and myalgia | 412 | 60.8 | 266 | 39.2 |
| Unlike the common cold, stuffy nose, runny nose, and sneezing are less common in persons infected with the COVID-19 virus | 633 | 93.4 | 45 | 6.6 |
| **Knowledge of high risk** | | | | |
| Not all persons with COVID-2019 will develop severe cases. Only those who are elderly, have chronic illnesses & are obese are more likely to develop severe cases | 468 | 69 | 210 | 31 |
| There currently is no effective cure for COVID-2019, but early symptomatic and supportive treatment can help most patients recover from the infection | 653 | 96.3 | 25 | 3.7 |
| **Knowledge about the mode of transmissions and infectiousness** | | | | |
| The COVID-19 virus spreads via respiratory droplets of infected individuals | 646 | 95.3 | 32 | 4.7 |
| Eating or contacting wild animals would result in infection by the COVID-19 virus | 609 | 89.8 | 69 | 10.2 |
| Persons with COVID-19 cannot infect the virus to others when a fever is not present | 568 | 83.8 | 110 | 16.2 |
| Proper washing hand with soap and water is one method of preventing COVID-19 | 370 | 54.6 | 308 | 45.4 |
| **Knowledge about ways of prevention** | | | | |
| One way of prevention of COVID 19 is not touching the eye, nose by unwashed hand | 624 | 92 | 54 | 8 |
| To prevent the infection by COVID-19, individuals should avoid going to crowded places such as train stations and avoid taking public transport | 640 | 94.4 | 38 | 5.6 |
| Ordinary residents can wear general medical masks to prevent the infection by the COVID-19 virus | 624 | 92 | 54 | 8 |
| People who have contact with someone infected with the COVID-19 virus should immediately be isolated to a proper place | 613 | 90.4 | 65 | 9.6 |
| Isolation and treatment of people who are infected with the COVID-19 virus are effective ways to reduce the spread of the virus | 77 | 11.4 | 601 | 88.6 |
| Children and young adults don't need to take measures to prevent the infection by the COVID-19 virus. | 518 | 76.4 | 160 | 13.6 |

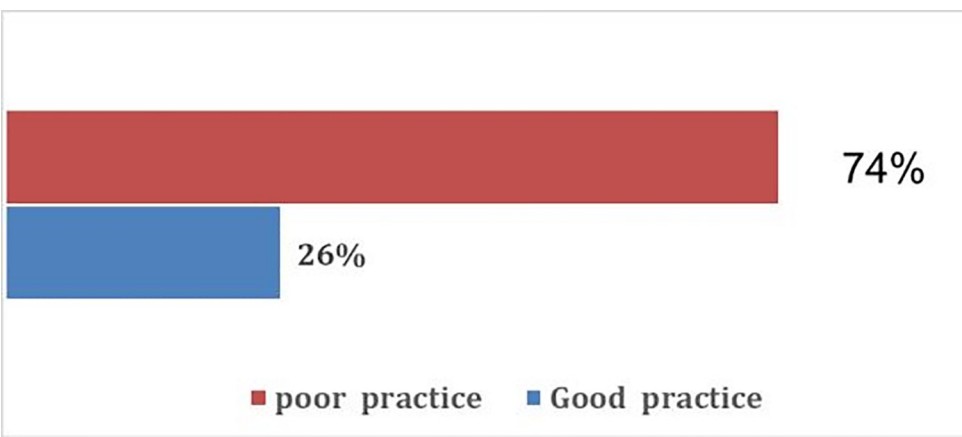

**Fig 4. Status of preventive practices towards COVID-19 among patients with hypertension or diabetes mellitus, Southern Ethiopia, July 2020.**

to COVID-19 infection. Health professionals were sources of information for the majority, 507 (55.2%) of respondents, followed by media 202 (22%). Five hundred and twenty-six 526 (77.6%) were diabetes patients. The majority, 648 (95.6%), had medical follow-up before the COVID-19 pandemic occurrence. Out of these, 509 (75.1%) had visited health facilities every month. However, during the COVID-19 pandemic, 192 (28.3%) had no follow-up at the health facilities. Out of those who discontinued follow-up, 144 (21.2%) were due to fear of acquiring COVID-19 infection from health facilities [**Table 4**].

## Adherence to drug and control measures among patients with hypertension or diabetes

Out of the 678 respondents, 550 (81.1%) had taken their drugs regularly. However, 128 (18.9%) respondents discontinued taking their drugs. The majority, 510 (75.2%), purchased their drugs once for three months. About 492 (72.6%) accessed it from public health facilities. Fifty-two (7.7%) conducted physical exercise as an alternative to the drug, and more than a half, 351(51.8%) understood taking of drugs properly and lifestyle modification as a means of controlling those specific diseases [**Table 5**].

**Table 3. Preventive practices towards COVID-19 among patients with hypertension or diabetes, Southern Ethiopia, July 2020.**

| Practice variables | Yes | | No | |
|---|---|---|---|---|
| | No. | % | No | % |
| Frequently wash hands with water and soap | 299 | 44.1 | 379 | 54.9 |
| Stopped shaking hands while giving greeting | 582 | 85.8 | 96 | 14.2 |
| Avoided proximity, including while greeting (within 2 m) | 624 | 92 | 54 | 8 |
| have not gone to a crowded place | 505 | 74.5 | 173 | 25.5 |
| Used face mask when leaving home | 433 | 63.9 | 245 | 36.1 |
| Avoid touching eyes, nose, and mouth before washing hands | 557 | 82.2 | 121 | 17.8 |
| Used cover /elbow during coughing/sneezing | 632 | 93.2 | 46 | 6.8 |
| Others (alcohol-rubbing, no contact with surfaces) | 468 | 64.6 | 240 | 35.4 |
| Have stayed at home | 440 | 64.9 | 238 | 35.1 |

**Table 4. Vulnerability and NCD follow up related characteristics among patients with hypertension or diabetes mellitus, Southern Ethiopia, July 2020.**

| Variables | Category | Frequency | Percent |
|---|---|---|---|
| Heard about COVID-19 | Yes | 666 | 98.2 |
| | No | 12 | 1.8 |
| Follow up before COVID-19 pandemic | Yes | 648 | 95.6 |
| | No | 30 | 4.4 |
| Reason not to follow up before COVID-19 pandemic | Lack of money | 16 | 2.4 |
| | I feel better | 14 | 2.1 |
| Frequency of follow up before COVID-19 pandemic | Every month | 509 | 75.1 |
| Follow up after COVID-19 pandemic | Every three month | 53 | 7.8 |
| | Every two month | 38 | 5.6 |
| Reason not to follow up after COVID-19 pandemic | When feeling sick | 48 | 7.1 |
| | Yes | 486 | 71.7 |
| | No | 192 | 28.3 |
| | Fear of acquiring COVID-19 | 144 | 21.2 |
| | I feel better | 39 | 5.8 |
| | Lack of money | 9 | 1.3 |

## Predictors of knowledge towards COVID-19 among study participants

Both bivariate and multivariate logistic regression analyses were done. In the bivariate analysis, the association of socio-demographic and other variables with the knowledge status of the respondents were checked; Family monthly income and medical follow-up after the COVID-19 pandemic were significantly associated at p <0.05. These variables which were significant in the bivariate analysis were entered into the multivariate logistic regression model. Therefore, family monthly income (AOR = 1.42; 95% CI: 1.04, 1.94) and medical follow-up after COVID-19 (AOR = 1.44; 95% CI: 1.02, 2.04) were found to be significant predictors of knowledge towards COVID-19 in this study. The odds of good knowledge among the study participants who had family monthly income above or equal to 2500 ETB had about 1.42 times

**Table 5. Adherence to drug and control measures among patients with hypertension or diabetes mellitus, Southern Ethiopia, July 2020.**

| Variables | Category | Frequency | Percent |
|---|---|---|---|
| Take drug regularly | Yes | 550 | 81.1 |
| | No | 128 | 18.9 |
| Where do you get the drug from | Public health facilities | 492 | 72.6 |
| | Private health facilities | 58 | 8.6 |
| Reason for discontinuation of drugs | Abscess of proper storage | 218 | 31.5 |
| | Lack of money | 56 | 8.1 |
| | I feel better | 52 | 7.5 |
| | Shifted to traditional medicines | 12 | 1.7 |
| What measures you have taken alternative to drug | Physical exercise | 52 | 7.7 |
| | Traditional medicines | 25 | 4.3 |
| | Spiritual remedy | 29 | 3.7 |
| | Nothing | 22 | 3.2 |
| How do you control hypertension/diabetes | Both proper taking of drugs & Lifestyle modifications | 351 | 51.8 |
| | Proper taking of drugs alone | 170 | 25.1 |
| | Lifestyle modifications alone | 157 | 23.2 |

**Table 6. Predictors of knowledge towards COVID-19 among patients with hypertension or diabetes mellitus, Southern Ethiopia, July 2020.**

| Variables | knowledge status | | COR (95%CI) | AOR (95%CI) |
|---|---|---|---|---|
| | **Good knowledge** | **Poor knowledge** | | |
| Religion | | | | |
| Orthodox | 300 (64.8%) | 163 (35.2%) | 1 | 1 |
| Protestant | 99 (70.2%) | 42 (29.8%) | 0.78 (0.51,1.17) | 0.82 (0.67,1.45) |
| Muslim | 43 (71.7%) | 17 (28.3%) | 0.72 (0.40, 1.31) | 0.65 (0.51, 1.47) |
| Other | 8 (57.1%) | 6 (42.9%) | 1.38 (0.47, 4.04) | 1.45 (0.53, 3.04) |
| Sex | | | | |
| Male | 280(64.2%) | 156 (35.8%) | 1 | 1 |
| Female | 170 (70.2%) | 72 (29.8%) | 1.31 (0.93,1.84) | 0.76 (0.54,1.06) |
| Marital status | | | | |
| Single | 111 (67.3%) | 54 (32.7%) | 1 | 1 |
| Married | 339 (66.1%) | 174 (33.9%) | 1.05 (0.72, 1.53) | 1.19 (0.81,1.71) |
| Family monthly income | | | | |
| <2500 | 202 (62.3%) | 122 (37.7%) | 1 | 1 |
| ≥2500 | 248 (70.1%) | 106 (29.9%) | 1.41 (1.02, 1.94) * | 1.42 (1.04, 1.94) ** |
| Educational status | | | | |
| No formal education | 75 (74.3%) | 26 (25.7%) | 0.62(0.36,1.05) | 0.58 (0.45, 1.35) |
| Can read and write | 42 (66.7%) | 21 (33.7%) | 0.89 (0.49, 1.62) | 0.72 (0.81, 1.52) |
| Primary education | 54 (60%) | 36 (40%) | 1.19 (0.72,1.98) | 1.21 (0.63,1.81) |
| Secondary education | 137 (67.5%) | 66 (32.5%) | 0.86 (0.57,1.29) | 0.8 (0.67,1.34) |
| Certificate and above | 142 (64.3%) | 79 (35.7%) | 1 | 1 |
| Follow up after COVID-19 | | | | |
| No | 136 (61%) | 87(39%) | 1 | 1 |
| Yes | 314 (69%) | 141 (31%) | 1.43 (1.02, 1.99) * | 1.44 (1.02, 2.04) ** |

** significant at p<0.01

* significant at P<0.05.

higher knowledge about COVID-19 than their counterparts. Respondents who had medical follow-up after COVID-19 had 1.44 times higher knowledge as compared to those who had no medical follow-up [Table 6].

## Predictors of preventive practices towards COVID-19 among patients with hypertension or diabetes mellitus

The predictors of preventive practices towards COVID-19 were assessed. Both bivariate and multivariate logistic regression analysis were done and those variables with P-value < 0.25 in the bivariate analysis were candidates for the final model. Therefore, the selected variables were entered into the multivariate logistic regression analysis using the 'backward stepwise' method of variable selection. In multivariate logistic regression, only variables such as knowledge status of the respondent (AOR = 1.47; 95% CI: 1.03, 2.12), follow up after COVID-19 (AOR = 2.21;95% CI:1.39, 3.52), and taking drugs regularly (AOR = 1.89; 95% CI:1.13, 3.17) were statistically significantly associated with preventive practices of COVID-19. The study participants who had good knowledge about COVID-19 were about 50% higher to have a good preventive practice as compared to those who had low knowledge. The likelihood of exercising preventive practices of COVID-19 was about two times higher among patients with hypertension/diabetes who had regular follow-up after COVID-19 than their counterparts [Table 7].

**Table 7. Predictors of preventive practices towards COVID-19 among patients with hypertension or diabetes mellitus, Southern Ethiopia, July 2020.**

| Variables | preventive practice status | | COR (95%CI) | AOR (95%CI) |
|---|---|---|---|---|
| | Good practice | Poor practice | | |
| Religion | | | | |
| Orthodox | 127(27.4%) | 336 (72.6.%) | 1 | 1 |
| Protestant | 31(22.0%) | 110 (78.0%) | 1.34 (0.85,2.09) | 1.52 (0.73,1.39) |
| Muslim | 15(25.0%) | 45 (75.0%) | 1. 13 (0.61, 2.10) | 1.34 (0.54, 2.46) |
| Other | 6 (42.9%) | 8 (57.1%) | 0.50 (0.17, 1.48) | 0.67 (0.42, 1.68) |
| Sex | | | | |
| Male | 116(26.6%) | 320 (73.4%) | 1 | 1 |
| Female | 63 (26.0%) | 179 (74.0%) | 1.03 (0.72,1.47) | 1.25 (0.54, 1.83) |
| Marital status | | | | |
| Single | 36(21.8%) | 129 (72.1%) | 1 | 1 |
| Married | 143 (27.9%) | 370 (72.1%) | 0.72 (0.47, 1.09) | 0.63 (0.25, 1.42) |
| Monthly income | | | | |
| <2500 | 86 (25.8%) | 247 (74.2%) | 1 | 1 |
| ≥2500 | 93 (27.0%) | 252 (73.0%) | 0.81 (0.58, 1.15) | 0.47 (0.76, 1.46) |
| Educational status | | | | |
| Cannot read and write | 27 (26.7%) | 74 (73.3%) | 0.93 (0.54, 1.58) | 0.58 (0.71, 1.43) |
| Can read and write | 17 (27.0%) | 46 (73.0%) | 0.91 (0.48, 1.73) | 0.74 (0.61, 1.41) |
| Primary education | 18 (20.0%) | 72 (80.0%) | 1.35 (0.74,2.47) | 1.25 (0.52, 2.51) |
| Secondary education | 61 (30.0%) | 142 (70.0%) | 0.79 (0.51,1.21) | 0.53 (0.43, 1.31) |
| Certificate and above | 56 (25.3%) | 162 (74.7%) | 1 | 1 |
| Follow up after COVID-19 | | | | |
| No | 84 (43.8%) | 108 (56.3%) | 1 | 1 |
| Yes | 95 (19.5%) | 391 (80.5%) | 2.02 (1.41, 2.89) * | 2.21 (1.39, 3.52) ** |
| Knowledge status of the respondent | | | | |
| Good knowledge | 113 (25.1%) | 337 (74.9%) | 1.55 (1.09, 2.19) * | 1.47 (1.03, 2.12) ** |
| Poor knowledge | 66 (28.9%) | 162 (71.1%) | 1 | 1 |
| take drugs regularly | | | | |
| No | 61 (47.7%) | 67 (52.3%) | 1 | 1 |
| Yes | 118 (21.8%) | 432 (78.5%) | 3.3 (2.22, 4.98) * | 1.89(1.13, 3.17) ** |

* * Significant at P <0.01

* Significant at P <0.05.

## Discussion

Pre-existing diabetes is a risk factor for poor outcomes and death after COVID-19. The association between COVID-19 and hyperglycemia in elderly patients with DM is likely to reflect metabolic inflammation and exaggerated cytokine release [21]. Recent data suggest that SARS-CoV2 infection can lead to a deterioration in glycemic control, involving both profound insulin resistance, and impaired insulin secretion, together with leading to diabetic ketoacidosis, DKA [22, 23]. Moreover, the impairment at different levels of the innate and adaptive immune response is likely to be involved in the poorer ability to fight infection in these patients with diabetes and contribute to severe forms of morbidity and mortality [24–26].

Hypertension is also one of the risk factors for disease severity and death from SARS-Cov2 infection. The potential biological mechanism is that hypertensive patients can be more prone to Renin-Angiotensin System (RAS) imbalance, which in turn lead to vasoconstriction/

inflammation due to unopposed Ang II effect [27]. This process is precipitated by increased Dipeptidyl Peptidase4 (DPP4) vascular activity/expression and by chronic low-grade inflammation [28]. The dysregulated response, allied with diminished physiologic cardiovascular reserve, induced by hypertension—arterial stiffening, left ventricular hypertrophy, and endothelial dysfunction creates the perfect milieu for both COVID-19 related tissue injury and worsening of cardiac, renal, and vascular function. This abnormal condition predisposes hypertensive patients to more complicated clinical outcomes [29].

Therefore, this study assessed the knowledge and preventive practices towards COVID-19 among patients with hypertension or diabetes mellitus in three large zones of Southern Ethiopia. The study revealed that 63% of patients with hypertension or diabetes mellitus had good knowledge about COVID-19 (Based on the knowledge score of the participants). This finding is consistent with a study conducted in Northwest, Ethiopia [30], in which the level of good knowledge towards COVID-19 among NCD patients was 66%. However, the current study finding is lower than studies conducted in China and Iran [20, 31], in which the overall achieved knowledge towards COVID-19 in both studies was 90%. The difference could be due to the socio-economic and demographic differences of these countries and ours.

This finding is higher than a study conducted in Thailand in which 73.4% of the study participants had poor knowledge of the pandemic [32]. This discrepancy may be, attributed to the time of study conducted. The current study was conducted when the number of cases alarmingly increased and the Federal Ministry of Health of Ethiopia has been engaged intensively in increasing the awareness of the general population. However, the study in Thailand was conducted during the time of early outbreak when the number of cases was very low.

This finding is lower than the overall knowledge towards COVID-19 among health care workers (HCWs) in China and Pakistan. The knowledge of HCWs towards COVID 19 prevention in both studies was 89 and 92.3% respectively [33, 34]. This discrepancy could be due to the difference in the study populations. The previous two studies were conducted among health care workers who have exposure to the information about the COVID-19 pandemic whereas the current study was conducted among NCD patients from the general population.

From specific knowledge assessing questions (knowledge about the mode of transmissions and infectiousness), 95% of the study subjects knew that the COVID-19 virus spreads via respiratory droplets of infected individuals. Concerning knowledge about the risk of vulnerability, 69% of respondents knew that not all persons with COVID-19 will develop severe cases, but only those who are elderly, have chronic illnesses & are obese are more likely to have severe forms of COVID-19 and subsequent mortality. This finding is in line with a study conducted in Jimma, Ethiopia [18].

The Ethiopian government has been working intensively on awareness creation towards this pandemic using different media, including social media, since March 2020, after a few new cases of COVID-19 reported in the capital city of the country. Henceforth, 98% of the study population heard about COVID-19 during the survey. This finding is similar to studies conducted in Ethiopia, a survey conducted in three countries in Africa, and Pakistan. In those studies, the number of study participants who heard about the disease accounts for 91.5, 94, and 90%, respectively [35–37].

In this study, only 26.4% of the respondents had good practice towards the prevention of COVID-19. Concerning the specific preventive practices, 55.1% of the respondents frequently washed their hands with soap and water, 64.6% used sanitizer or alcohol rubs. There was a significant gap between knowledge and preventive practices towards the pandemic among the study subjects. For instance, the knowledge of wearing medical masks to prevent infection was 92%, but the practice of wearing medical masks was only 63.9%. Knowledge of avoiding going to crowded places was 94.4%, whereas the practice of going to crowded places was 74.5%. This

finding is consistent with studies conducted in Ethiopia [18, 30]. However, this result is inconsistent with studies conducted in India and Bangladesh, in which above 90% of the study subjects avoided crowded places and wore face masks when leaving home during the rapid rise period of the COVID-19 outbreak [38, 39]. This discrepancy may be due to socio-cultural and behavioral differences. Ethiopia is known for its diversity, social, cultural, religious ceremonies, and a high rate of overcrowded living conditions.

Moreover, financial, cultural, and religious norms are the main barriers to preventive practices of COVID-19 that affect the acceptability of public health measures [40, 41]. In this study, the high practice of going to crowded places during the pandemic is attributed to the strong cultural and religious norms. These norms are the restricting factors of preventive practices of COVID-19. In Ethiopian culture, practicing social gatherings during the occurrence of vital events is common. All rituals, including mourning, marriage, and other social and religious gatherings are long-lasting practices persisting in the era of COVID 19. This compromises the acceptance of the public health measures by the community which further challenges the effective implementation of the measures against the COVID-19 pandemic.

Family income significantly predicted knowledge of COVID-19. The odds of having good knowledge among the study participants who had family income of more than or equal to 2500 ETB per month was about 42 percent higher than their counterparts. This finding is supported by other studies conducted in Ethiopia, China, Malaysia, & America [18, 37, 39, 42–44], where high income was associated with good knowledge about COVID-19. The potential reason could be attributed to the economic status, which has a significant influence on the change of human health behavior. Moreover, the low economic status of people can hinder the ability to cover costs related to personal protective materials like face masks and others for daily consumption.

Positive attitudes and preventive practices towards COVID-19 are modified by knowledge through successive works on increasing awareness and change of behavior [30]. The risk of infection with COVID-19 decreases by improving knowledge about the disease and patients' preventive practices [45]. In this study, Knowledge about COVID-19 was significantly associated with preventive practices of COVID-19. Respondents who had good knowledge about COVID-19 were more likely to exercise the preventive practices towards COVID-19 than their counterparts. This finding is consistent with studies conducted in Ethiopia and China [18, 30, 31].

A study conducted in Addis Ababa city administration of Ethiopia showed that 40–60% of NCD patients discontinued their regular medical follow-up during the first week after notification of the first positive COVID-19 case in the country [46]. In this study, 28.2% of patients discontinued their regular medical follow-up at health facilities and 18.9% of patients did not take their medications regularly during the COVID-19 pandemic. Fear of acquiring COVID-19 infection was the most frequent reason for discontinuation of their medical follow-up. In the previous study, the discontinuation rate was higher than in the current study. the reason for this discrepancy is that the previous study was conducted in the earlier period of the pandemic, the fear and frustration towards the disease was initially higher.

## Limitations

This study is limited by its cross-section nature, whereby causal inferences may not be established; this limits the interpretation of the estimated associations. Moreover, this study should have assessed the attitudes and perceptions of patients through in-depth interviews and constructed them as multi-dimensional measures. Thus, the findings of this study should be interpreted within these limitations.

## Conclusions

In this study, the knowledge of the respondents towards the COVID-19 pandemic was good. But the preventive practice was very low. There was a significant gap between knowledge and preventive practices towards the COVID-9 pandemic among the study subjects. Monthly income was significantly associated with knowledge of COVID-19. Knowledge of COVID-19 was found to be an independent predictor of preventive practices towards COVID-19. Health education programs aimed at mobilizing and improving COVID-19- related knowledge and practices are highly recommended for these patients with hypertension or diabetes mellitus. Concerning the preventive practices, great emphasis should be given to specific preventive practices such as frequent handwashing with soap and water, avoiding going to crowded places (social distancing), and using face masks while leaving home. These major preventive practices have to be adopted to prevent the contraction of the virus. Furthermore, for fear of acquiring the disease from health facilities, a significant number of patients with hypertension or diabetes interrupted their medical follow-up. Back tracing of those patients at a community level, continuing to follow up, a priority of testing, and vaccinations against COVID-19 at home level is highly recommended for patients with hypertension or diabetes mellitus.

## Supporting information

**S1 File. Dataset of NCDs and vulnerability to COVID-19: The case of adult patients with hypertension or diabetes mellitus in Gamo, Gofa, and South Omo zones in Southern Ethiopia.**
(XLSX)

**S2 File. English and Amharic version questionnaires.**
(ZIP)

## Acknowledgments

We would like to thank Arba Minch College of Health Sciences for taking the initiative and financial support to undertake this research. Our heartfelt thanks also go to the CEOs of Arba Minch, Sawla, and Jinka General Hospitals for their willingness and support during the survey. Last but not the least, we would like to pass our gratitude to the study participants for their willingness and giving time to complete the interview.

## Author Contributions

**Conceptualization:** Fikre Bojola, Wondimagegn Taye, Bahiru Mulatu, Aleme Mekuria.

**Data curation:** Habtamu Samuel, Bahiru Mulatu, Aknaw Kawza, Aleme Mekuria.

**Formal analysis:** Fikre Bojola, Bahiru Mulatu, Aleme Mekuria.

**Funding acquisition:** Wondimagegn Taye, Aknaw Kawza.

**Investigation:** Fikre Bojola, Bahiru Mulatu, Aleme Mekuria.

**Methodology:** Fikre Bojola, Bahiru Mulatu, Aleme Mekuria.

**Project administration:** Wondimagegn Taye, Habtamu Samuel.

**Resources:** Wondimagegn Taye.

**Software:** Fikre Bojola, Aleme Mekuria.

**Supervision:** Wondimagegn Taye, Habtamu Samuel, Aknaw Kawza, Aleme Mekuria.

**Validation:** Fikre Bojola, Habtamu Samuel, Bahiru Mulatu, Aleme Mekuria.

**Visualization:** Wondimagegn Taye, Habtamu Samuel, Bahiru Mulatu, Aknaw Kawza.

**Writing – original draft:** Aleme Mekuria.

**Writing – review & editing:** Fikre Bojola, Bahiru Mulatu, Aleme Mekuria.

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
