## [Decision Letter · Decision Letter 0]

27 May 2021

PONE-D-20-31282

Chronic diseases and vulnerability to COVID-19: the case of adult people with Chronic diseases in Gamo, Gofa, and South Omo zones in Southern Ethiopia

PLOS ONE

Dear Dr. Mekuria,

Thank you for submitting your manuscript to PLOS ONE. After careful consideration, we feel that it has merit but does not fully meet PLOS ONE’s publication criteria as it currently stands. Therefore, we invite you to submit a revised version of the manuscript that addresses the points raised during the review process.

We look forward to receiving your revised manuscript.

Kind regards,

Tauqeer Hussain Mallhi, Ph.D

Academic Editor

PLOS ONE

Journal Requirements:

2. Please include additional information regarding the survey or questionnaire used in the study and ensure that you have provided sufficient details that others could replicate the analyses. For instance, if you developed a questionnaire as part of this study and it is not under a copyright more restrictive than CC-BY, please include a copy, in both the original language and English, as Supporting Information.  If the original language is written in non-Latin characters, for example Amharic, Chinese, or Korean, please use a file format that ensures these characters are visible.

Additional Editor Comments:

Dear Authors, thank you for submitting in Plos One. Your manuscript has been assessed by relevant experts from the field. They found manuscript interesting but raised substantial concerns in methodology (selection of participants, operational definitions) and interpretation of results. Moreover, reviewers raised issues on bias and generalizability of the findings. It is requested to please consider the comments of reviewers.

Reviewers' comments:

Reviewer's Responses to Questions

**Comments to the Author**

1. Is the manuscript technically sound, and do the data support the conclusions?

Reviewer #1: Yes

Reviewer #2: Yes

Reviewer #3: Partly

Reviewer #4: Yes

2. Has the statistical analysis been performed appropriately and rigorously? 

Reviewer #1: Yes

Reviewer #2: I Don't Know

Reviewer #3: Yes

Reviewer #4: Yes

3. Have the authors made all data underlying the findings in their manuscript fully available?

Reviewer #1: Yes

Reviewer #2: Yes

Reviewer #3: Yes

Reviewer #4: Yes

4. Is the manuscript presented in an intelligible fashion and written in standard English?

Reviewer #1: No

Reviewer #2: Yes

Reviewer #3: No

Reviewer #4: Yes

5. Review Comments to the Author

Reviewer #1: Identifying the sociodemographic factors influencing the preventive behavior is important for defeating the pandemic of COVID-19. In this manuscript, Fikre Bojola et al assessed the association between sociodemographic factors and the knowledge or behavior on COVID-19 prevention. The study collected data from patients with chronic diseases in southern Ethiopia, using a structured questionnaire and a community based cross-sectional study design. The authors found that many sociodemographic factors have influences and more striking is the knowledge and behavior showed big discrepancy. The study done here has a good starting point, however further analyses and in depth discussions are still needed.

Major concerns:

(1) The title of this manuscript is not reflecting the actual purpose and research type of this study.

(2) Since the knowledge and behavior showed huge discrepancy, find out the key rescrictions on preventive practice and find ways to change the behavior would be of great value. The key restriction factors need to be deeply analyzed and summarized.

(3) It would be helpful to compare the key restriction factors of preventive practice identified in this study with key restrictions identified from previous sociodemographic studies conducted in other region or coundries around the wrold.

Minor concerns:

(1) Most of the results are presented in the form of tables, which will greatly enhance the readability if represented by graphs, especially for the significantly associated factors.

(2) The abbreviations and meaning of the numbers in the Supplemental dataset should be described.

(3) The text needs reviewing by a professional English editor as many sentences are still ambiguous.

Reviewer #2: GENERAL COMMENTS

This report shows the gap between knowledge about COVID-19 and preventive practices among patients with hypertension or diabetes in three regions of Ethiopia. Despite the months that have passed since the outbreak of the pandemic, the frequency of preventive practices among subjects who are high-risk patients was low. This may be explained by the regional characteristics and low income. There are several similar studies, but this regionalism is what makes this paper unique. This study suggests that NCDs patients are the target population for health education programs on COVID-19 prevention.

SPECIFIC COMMENTS

Major

i)Line129: The reasons for choosing hypertension and diabetes as representatives of NCDs are not presented.

It is recommended to indicate the percentage of patients receiving medication.

ii)Line162,166: Why did you set the cutoff point for each score at 60%? Showing a histogram of the scores is a good way to interpret the results.

iii)Table-1: What is your rationale for dividing income into two groups at 2500ETB?

Do you have data on body mass index or obesity rate? Obesity is an important risk factor for the severity of COVID-19, which is also associated with lifestyle and NCDs.

iv)The limitations of the study are not mentioned in this report.

Selection bias exists because only hypertension or diabetes were selected as NCDs. And it is also desirable to describe the bias inherent in the questionnaire method.

Minor

i)Line36:It is suitable to describe “patients with hypertension or diabetes mellitus" than "chronic patients".

ii)Line192: Please add some more information about the bivariate analysis method.

iii)Table-1: It is recommended that continuous variables (Age and Income) show mean or median values. Or could you provide histograms of them?

Reviewer #3: Overall the authors have attempted to survey a cohort of patients with NCDs to assess knowledge and perceptions of COVID-19 in the southern regions of Ethiopia. I acknowledge all the hard the work that went into the construction and implementation of the study but much is needed to improve the manuscript (in my opinion) if acceptable for publication in PLOS ONE.

I have some suggestions moving forward:

- Many grammatical and sentence structure errors exist in the manuscript and should be edited accordingly; manuscript in my opinion is not acceptable at current stage of standard English being presented.

- Title is misleading. It appears the participants were selected from those who only had hypertension and diabetes mellitus. Were those with other NCDs (if present) included? Will need a major revision to focus on those with hypertension and diabetes.

- Fatality rates vary on region of the world and other factors, so I have a hard time accepting "overall fatality of COVID-19 is low" which is described multiple times in the manuscript, including the abstract. Please rephrase this and add some statistics relevant to Ethiopia known fatality rate

- Line 29 needs rewording. I think you are trying to say that those with NCDs are more likely to be non-adherent to medications and other life style related recommendations during COVID-19

- throughout the manuscript you use the term "chronic patients" and this needs to be replaced something else that is more clear, such as "among those with NCDs".

- In the abstract please redesign and leave out what software was used to conduct the analysis. How were these patients selected and in what setting?

- Methods: please include a map of the three zones which will help the readers understand the geography of Ethiopia and the where the study took place

- Results: 96.3% response rate? that seems too high to be reliable. How did they come to that conclusion. Need describe engagement protocol for introducing the research project to a potential participant.

- Why the split in age above and below 30 years? I recommend dividing into 18-30, 31-50, 51-64, 65-80, 81+

- line 264: "bad practices" please explain differently

Adherence to drug and control measures NCDs: Does this only pertain to hypertension and diabetes medications? Insulin therapy?

- Discussion: needs total revamping; will need to focus on diabetes and hypertension among COVID-19 and cannot say "chronic patients" and NCDs, as it is unclear if others with NCDs were included.

- Does not have limitations to the study and there are several

- Line 414: why is "almost" used here and also found in the abstract

Reviewer #4: The presented descriptive study by Bojola et al. has aimed to address the multi-dimensional knowledge about the COVID-19 spread and the preventive practices followed by the chronic disease subjects, who are vulnerable to COVID-19 infection, in the three selected zones of Ethiopia. Such community-based studies are crucial to create awareness and to identify the factors that require special attention to prevent and/or control the spread of COVID-19 infection. Interestingly the data presented in this study reflect the importance of knowledge, in addition to monthly income, in practicing preventive behaviors among these study participants. Establishing again knowledge is the key to good practice. Although the data has been presented in a suitable format, the background information provided in the Introduction is hard to verify from the given references. It would be better to give an appropriate reference at the end of every claim, but not at the end of the paragraph. Particularly, the previous study results mentioned for ref # 8 through #13 have become difficult to cross-check. Further, multiple references are not in the right format and many of them with missing information (volume and page numbers).

6. PLOS authors have the option to publish the peer review history of their article (what does this mean?). If published, this will include your full peer review and any attached files.

Reviewer #1: No

Reviewer #2: No

Reviewer #3: No

Reviewer #4: No

---

## [Author Response · Author response to Decision Letter 0]

1 Jul 2021

Response to the editor’s and reviewer’s comments

Dear Editor/Reviewers, we would like to thank the editor and reviewers for giving their time to review the manuscript. We found the reviewers’ comments/feedback very helpful in improving the manuscript and we have revised the manuscript accordingly. Please find attached the revised manuscript. Our point-by-point descriptions on the suggested revisions are below. 

Kind regards,

Aleme Mekuria 

Reviewer #1: 

Comment 1

Identifying the socio demographic factors influencing the preventive behavior is important for defeating the pandemic of COVID-19. In this manuscript, Fikre Bojola et al assessed the association between sociodemographic factors and the knowledge or behavior on COVID-19 prevention. The study collected data from patients with chronic diseases in southern Ethiopia, using a structured questionnaire and a community based cross-sectional study design. The authors found that many socio demographic factors have influences and more striking is the knowledge and behavior showed big discrepancy. The study done here has a good starting point, however further analyses and in-depth discussions are still needed.

Response: Authors express gratitude to the reviewer for the appreciation. Of course, the study revealed a huge gap between knowledge and preventive practices towards the pandemic among the study subjects. Moreover, we forwarded recommendations for the concerned bodies to use this finding as a baseline for further studies and undertake community mobilization interventions towards these vulnerable community groups.

Comment

(1) The title of this manuscript is not reflecting the actual purpose and research type of this study.

Response: 

Thank you. Since the study is about patients with hypertension and Diabetes Mellitus, we have made modification on the title to show the study subjects are these particular NCDs.

Comment:

(2) Since the knowledge and behavior showed huge discrepancy, find out the key restrictions on preventive practice and find ways to change the behavior would be of great value. The key restriction factors need to be deeply analyzed and summarized.

(3) It would be helpful to compare the key restriction factors of preventive practice identified in this study with key restrictions identified from previous sociodemographic studies conducted in other region or countries around the world.

Response 

Thank you. We have identified and discussed the key restriction factors on preventive practices of COVID-19 comparing with other findings. Please kindly find in the discussion section page 20, line 394-408.

Comment

(1) Most of the results are presented in the form of tables, which will greatly enhance the readability if represented by graphs, especially for the significantly associated factors.

Response: 

Thank you for the comment. In the revised version of the manuscript, we displayed the level of the multidimensional knowledge (MDK) and the level of preventive practices towards COVID-19 analysis results in the form of pie and bar charts respectively. The figures captions are in the text but the figures are separately uploaded as per the PLose One manuscript submission guideline. Please kindly find on page 10, line 237-238 and page 12, line 272-273

Comment:

 The text needs reviewing by a professional English editor as many sentences are still ambiguous

Response:

We corrected/edited grammatical and sentence structure errors across the document. We used grammar editing software, “Grammarly” online software to edit the spelling, grammar and language usage. Finally, we used a native English language editor to check for the grammar and sentence structure errors.

Reviewer #2: 

GENERAL COMMENTS:

This report shows the gap between knowledge about COVID-19 and preventive practices among patients with hypertension or diabetes in three regions of Ethiopia. Despite the months that have passed since the outbreak of the pandemic, the frequency of preventive practices among subjects who are high-risk patients was low. This may be explained by the regional characteristics and low income. There are several similar studies, but this regionalism is what makes this paper unique. This study suggests that NCDs patients are the target population for health education programs on COVID-19 prevention.

Response: Authors express gratitude to the reviewer for his/her appreciation.

Comment:

Line129: The reasons for choosing hypertension and diabetes as representatives of NCDs are not presented.

Response: 

Thank you. We have stated the reason for choosing hypertension and diabetes mellitus patients in the revised version of the manuscript. Please kindly check page 5, under “methods and materials section”, population and sample (sub-section). Line 137-143

Comment:

It is recommended to indicate the percentage of patients receiving medication.

Response:

Thank you for the comment. We have already indicated the number and percentage of patients who are receiving their medication regularly and these who discontinued medication in table 5 and in the form of text above this table. Please kindly refer on page 14. Line 292-293

Comment:

Line162,166: Why did you set the cutoff point for each score at 60%? Showing a histogram of the scores is a good way to interpret the results.

Response:

Authors express gratitude to the reviewer for the comment. We used the bloom’s cut-off points during analysis for both outcome variables (knowledge and preventive practice towards COVID-19) as ‘good’ if the score was between 80 and 100% (11-14 points), ‘moderate’ if the score was between 60 and 79% (9-10 points), and ‘poor’ if the score was less than 60% (<8 points). For analysis, we recoded the outcome variables in to two (60% and above, ‘good’ and below 60%, ‘poor’).

Comment:

Table-1: What is your rationale for dividing income into two groups at 2500ETB?

Response: 

Thank you for the question raised. We categorized the income rate based on the indicator of proportion of population below the international poverty line, which is defined as the percentage of the population living on less than $1.90 a day at international prices. The 'international poverty line' is currently set at $1.90 a day at international prices. We computed the income variable by multiplying $1.90 by the current exchange rate (Ethiopian Birr currency). Therefore, the cut-off point is 2500 ETB per month. Meaning <2500 (below poverty line) and >2500 (above poverty line).

Comment:

Do you have data on body mass index or obesity rate? Obesity is an important risk factor for the severity of COVID-19, which is also associated with lifestyle and NCDs.

Response: 

Thank you! obesity is one of the most important risk factors for different kinds of NCDs like diabetes mellitus, cardiovascular diseases like hypertension, coronary artery diseases (CAD) and others. People with those diseases are at risk for the severity of COVID-19. Obesity may also be a risk factor for severity of COVID-19. In our case, we didn’t consider BMI when developing the tool. 

Comment:

The limitations of the study are not mentioned in this report.

Response:

Authors express thanks for the comment. We included the limitations of the study in the revised manuscript (just above the conclusion section). Please kindly find it on page 21, line 422-426

Comment 

Selection bias exists because only hypertension or diabetes were selected as NCDs. And it is also desirable to describe the bias inherent in the questionnaire method.

Response:

Thank you for the comment. Because of the insignificant number of other NCD cases in the study area, we collected the data only from hypertensive and diabetic patients. The data and the conclusion drawn from the study goes to patients with hypertension and diabetes mellitus rather all NCDs. Therefore, we made changes on the words “chronic patients” and “NCDs” and replaced with these specific NCDs (Hypertension and diabetes mellitus).

Comment:

 Line36: It is suitable to describe “patients with hypertension or diabetes mellitus" than "chronic patients".

Response: 

Thank you, line 34, “chronic patients” is replaced with “patients with hypertension or diabetes mellitus". Now this change is made consistent across the document in the revised manuscript. 

Comment

Line192: Please add some more information about the bivariate analysis method.

Response:

We described in detail about the bivariate and multivariate analysis and rephrased the paragraph starting from line 202-211, page 8.

Comment:

Table-1: It is recommended that continuous variables (Age and Income) show mean or median values. Or could you provide histograms of them?

Response: 

Thank you. The mean value of age and the median value of income is presented in text form in the first paragraph under the result section (just above the first table). Line 222 and 226, page 8.

Reviewer #3: 

Comment:

Overall, the authors have attempted to survey a cohort of patients with NCDs to assess knowledge and perceptions of COVID-19 in the southern regions of Ethiopia. I acknowledge all the hard the work that went into the construction and implementation of the study but much is needed to improve the manuscript (in my opinion) if acceptable for publication in PLOS ONE.

I have some suggestions moving forward:

Comment:

Many grammatical and sentence structure errors exist in the manuscript and should be edited accordingly; manuscript in my opinion is not acceptable at current stage of standard English being presented.

Response: 

We corrected/edited grammatical and sentence structure errors across the document. We used grammar editing software, “Grammarly” online software to edit the spelling, grammar and language usage. Finally, we used a native English language editor to check for any typographical or grammatical errors.

Comment 

Title is misleading. It appears the participants were selected from those who only had hypertension and diabetes mellitus. Were those with other NCDs (if present) included? Will need a major revision to focus on those with hypertension and diabetes.

Response: 

Due to the insignificant number of other NCDs, the study was conducted among those people with hypertension or diabetes. Others with NCDs were not included. Therefore, conclusions were drawn based on these patients. Moreover, we revised the entire document focusing on hypertensive and diabetic patients. Some modifications were also made on the title.

Comment:

Fatality rates vary on region of the world and other factors, so I have a hard time accepting "overall fatality of COVID-19 is low" which is described multiple times in the manuscript, including the abstract. Please rephrase this and add some statistics relevant to Ethiopia known fatality rate

 Response: 

We have rephrased the sentences in the abstract (line 27-30) and under the “introduction” section (line 74-79), page 3

Comment:

- Line 29 needs rewording. I think you are trying to say that those with NCDs are more likely to be non-adherent to medications and other life style related recommendations during COVID-19

Response: 

We have rephrased the first paragraph under the abstract section (27-30)

Comment:

- throughout the manuscript you use the term "chronic patients" and this needs to be replaced something else that is clearer, such as "among those with NCDs".

Response: 

 Thank you for the issue raised. We have checked the technical words used in this manuscript. We have replaced the word “chronic patients” with “patients with hypertension or diabetes” in the revised manuscript starting from the title page. 

Comment:

- In the abstract please redesign and leave out what software was used to conduct the analysis. How were these patients selected and in what setting?

Response: we rephrased and made corrections in the abstract under the ‘method’ section

Comment:

- Methods: please include a map of the three zones which will help the readers understand the geography of Ethiopia and the where the study took place

Response:

Thank you, we have now included the map of the country and the three zones where the study took place under the ‘methods’ section. The legend of the map is in the main text but the figure is uploaded separately as per the Plose One manuscript submission guideline. Please kindly check line 128-129, page 5.

Comment:

Results: 96.3% response rate? that seems too high to be reliable. How did they come to that conclusion? Need describe engagement protocol for introducing the research project to a potential participant.

Response: 

We used different strategies to achieve the maximum response rate; after sampling the study participants and before data collection, we made confirmations of the selected individuals for their presence by making home to home assessment using local selected guides. For those individuals who were not alive or changed residence, we included the next individual from the same register. During the data collection time, we made up to three visits for an individual who was not present at home during the time of visit. During sample size calculation, we added 10% of the calculated sample size considering non- response rate. For these reasons mentioned, we achieved the 96.3% response rate.

Comment:

- Why the split in age above and below 30 years? I recommend dividing into 18-30, 31-50, 51-64, 65-80, 81+

Response:

 Thank you for the comment. We re-categorized the age category in the revised manuscript (please kindly have a look the first paragraph (line 221-222) and the first part of the socio-demographic characteristics table (page 9).

Comment:

line 264: "bad practices" please explain differently

Response: The term “bad practices” is replaced with “poor practice” in the revised manuscript.

Comment: 

Adherence to drug and control measures NCDs: Does this only pertain to hypertension and diabetes medications? Insulin therapy?

Response: 

“Adherence to drug and control measures NCDs” is re-written to “Adherence to drug and control measures towards hypertension and diabetes mellitus.” Moreover, we have revised similar texts across the document to indicate the study subjects are patients with hypertension or diabetes mellitus. Line 292 & 299.

Comment:

Discussion: needs total revamping; will need to focus on diabetes and hypertension among COVID-19 and cannot say "chronic patients" and NCDs, as it is unclear if others with NCDs were included.

Response: 

Thank you for your comment. Data is collected only from those patients with hypertension or diabetic mellitus. Therefore, we made changes and rephrased sentences in the revised version. Instead of using “chronic patients" and NCDs, we used patients with hypertension or diabetes mellitus throughout the document including the discussion section. 

Comment:

- Does not have limitations to the study and there are several

Response:

Thank you for the comment. The study has limitations. It was not included previously. We now included the limitation in the revised manuscript just above the “conclusion” section. Line 424-428 page 21.

Comment:

- Line 414: why is "almost" used here and also found in the abstract

Response:

 The term "almost" is removed from line 414 and from the abstract

Reviewer #4: 

The presented descriptive study by Bojola et al. has aimed to address the multi-dimensional knowledge about the COVID-19 spread and the preventive practices followed by the chronic disease subjects, who are vulnerable to COVID-19 infection, in the three selected zones of Ethiopia. Such community-based studies are crucial to create awareness and to identify the factors that require special attention to prevent and/or control the spread of COVID-19 infection. Interestingly the data presented in this study reflect the importance of knowledge, in addition to monthly income, in practicing preventive behaviors among these study participants. Establishing again knowledge is the key to good practice. 

Comment:

Although the data has been presented in a suitable format, the background information provided in the Introduction is hard to verify from the given references. It would be better to give an appropriate reference at the end of every claim, but not at the end of the paragraph. Particularly, the previous study results mentioned for ref # 8 through #13 have become difficult to cross-check. Further, multiple references are not in the right format and many of them with missing information (volume and page numbers).

Response:

Authors are thankful to the reviewer for raising the issue. We have rearranged the references based on the given comments. The text has been revised now. References #8 through # 13 have been revised. Other references are now consistent and written in the right format (missing information are fulfilled. Please kindly check the reference section.

Thank you all for your valuable comments!

---

## [Decision Letter · Decision Letter 1]

27 Jul 2021

PONE-D-20-31282R1

Non-communicable diseases (NCDs) and Vulnerability to COVID-19: the case of adult patients with Hypertension or Diabetes Mellitus in Gamo, Gofa and South Omo zones in Southern Ethiopia

PLOS ONE

Dear Dr. Mekuria,

Thank you for submitting your manuscript to PLOS ONE. After careful consideration, we feel that it has merit but does not fully meet PLOS ONE’s publication criteria as it currently stands. Therefore, we invite you to submit a revised version of the manuscript that addresses the points raised during the review process.

We look forward to receiving your revised manuscript.

Kind regards,

Tauqeer Hussain Mallhi, Ph.D

Academic Editor

PLOS ONE

Journal Requirements:

Additional Editor Comments (if provided):

Thank you for revising the draft and addressing the comments of the reviewers. However, referees found few minor errors in the draft which should be considered before taking any decision for the manuscript. Please consider the comments of the reviewers and send us revised version at your convinience.

Reviewers' comments:

Reviewer's Responses to Questions

**Comments to the Author**

1. If the authors have adequately addressed your comments raised in a previous round of review and you feel that this manuscript is now acceptable for publication, you may indicate that here to bypass the “Comments to the Author” section, enter your conflict of interest statement in the “Confidential to Editor” section, and submit your "Accept" recommendation.

Reviewer #1: (No Response)

Reviewer #2: (No Response)

Reviewer #3: All comments have been addressed

Reviewer #4: All comments have been addressed

2. Is the manuscript technically sound, and do the data support the conclusions?

Reviewer #1: Yes

Reviewer #2: Yes

Reviewer #3: Yes

Reviewer #4: Yes

3. Has the statistical analysis been performed appropriately and rigorously? 

Reviewer #1: No

Reviewer #2: I Don't Know

Reviewer #3: Yes

Reviewer #4: Yes

4. Have the authors made all data underlying the findings in their manuscript fully available?

Reviewer #1: No

Reviewer #2: Yes

Reviewer #3: Yes

Reviewer #4: Yes

5. Is the manuscript presented in an intelligible fashion and written in standard English?

Reviewer #1: No

Reviewer #2: Yes

Reviewer #3: Yes

Reviewer #4: Yes

6. Review Comments to the Author

Reviewer #1: The authors declared, Knowledge of COVID-19 was found to be an independent 433 predictor of preventive practices towards COVID-19. Health education programs aimed at mobilizing 434 and improving COVID-19- related knowledge and practices are highly recommended for these patients 435 with hypertension or diabetes mellitus.

No independent data is sue to replicate this finding, I suggest to use data from other countries to verify the finding.

The potential biological mechanisms should be deeply discussed.

Reviewer #2: COMMENTS

This paper has been corrected for format and grammatical errors.

But there are still a few things that need to be fixed.

Line87-91:When you are going to address the relationship between COVID-19 and body mass index(BMI), it is appropriate to mention BMI in the discussion. If you were unable to obtain data on BMI, it is better to specify it.

Line225:It is preferable to include in the text the reason for setting the cutoff to 2500 ETB.

I recommend to format each tables properly;The number of the table, alignments, and typographical errors.

Reviewer #3: Authors have taken into consideration all comments given by the reviewers. They have addressed these recommendations and have provided a much improved manuscript.

Reviewer #4: The revised manuscript submitted by Bojola et al shows the improvements in data presentation and has also addressed the reviewer's comments appropriately. This study addresses an important issue of the community to prevent the spread of COVID-19 infection. Overall, with the survey conducted among non-communicable disease subjects from Ethiopia, who are vulnerable to COVID-19 infection, this study clearly shows the importance of knowledge and preventive practices to control/prevent COVID-19 spread. However, there are minor typos in the manuscript that can be corrected during the publication process.

7. PLOS authors have the option to publish the peer review history of their article (what does this mean?). If published, this will include your full peer review and any attached files.

Reviewer #1: No

Reviewer #2: **Yes: **Yuuki Bamba

Reviewer #3: **Yes: **Norman Beatty, MD, University of Florida College of Medicine, Gainesville, Florida, USA

Reviewer #4: No

---

## [Author Response · Author response to Decision Letter 1]

19 Aug 2021

Response to the editor’s and reviewer’s comments

Dear Editor/Reviewers, we are very glad to have your valuable comments. We have now revised the manuscript based on your comments.

Kind regards,

Aleme Mekuria 

Reviewer #1: 

Thank you for the important point you raised. 

We used data from other countries to verify the finding and discuss the potential biological mechanisms. Therefore, 

1. We deeply discussed the biological and pathophysiological mechanisms of both diabetes and hypertension 

2. The relationship of these diseases with COVID-19, and 

3. How these diseases worsen the clinical outcomes of COVID-19

4. We used nine additional references from other countries to discuss the above points 

Please kindly refer to the discussion section of the revised manuscript on page 20, line number 372-388 (the first two paragraphs) and the reference section (ref. no 22-30).

Reviewer # 2

1. We have corrected/edited grammatical and sentence structure errors across the document using online grammar editing software to edit the spelling, grammar, and language usage. Additionally, we used a professional English language editor to check for grammar and sentence structure errors. Therefore, we provided a much-improved manuscript.

2. We have included the reason for setting the cut-off point, 2500 ETB for the monthly income variable in the document. Please kindly refer to page 8, line 220-228.

3. We have corrected each table for the format, alignments, and typographical errors.

Thank you all for your commitment to evaluate our manuscript!

---

## [Decision Letter · Decision Letter 2]

13 Sep 2021

PONE-D-20-31282R2Non-communicable diseases (NCDs) and Vulnerability to COVID-19: the case of adult patients with Hypertension or Diabetes Mellitus in Gamo, Gofa, and South Omo zones in Southern EthiopiaPLOS ONE

Dear Dr. Mekuria,

Thank you for submitting your manuscript to PLOS ONE. After careful consideration, we feel that it has merit but does not fully meet PLOS ONE’s publication criteria as it currently stands. Therefore, we invite you to submit a revised version of the manuscript that addresses the points raised during the review process.

We look forward to receiving your revised manuscript.

Kind regards,

Tauqeer Hussain Mallhi, Ph.D

Academic Editor

PLOS ONE

Additional Editor Comments (if provided):

Dear Authors, thank you for submitting in Plos One. Your manuscript has been re-assessed by relevant experts from the field. They found manuscript interesting but raised some more concerns while discussing the findings. It is requested to please consider the comments of reviewer. It must be noted that reviewer has referred few citations in the comments. You are free to select whether these references fit to your discussion or not. We don't encourage any coercive citations.

Reviewers' comments:

Reviewer's Responses to Questions

**Comments to the Author**

1. If the authors have adequately addressed your comments raised in a previous round of review and you feel that this manuscript is now acceptable for publication, you may indicate that here to bypass the “Comments to the Author” section, enter your conflict of interest statement in the “Confidential to Editor” section, and submit your "Accept" recommendation.

Reviewer #1: (No Response)

Reviewer #2: All comments have been addressed

Reviewer #3: All comments have been addressed

Reviewer #4: All comments have been addressed

2. Is the manuscript technically sound, and do the data support the conclusions?

Reviewer #1: Yes

Reviewer #2: Yes

Reviewer #3: (No Response)

Reviewer #4: Yes

3. Has the statistical analysis been performed appropriately and rigorously? 

Reviewer #1: No

Reviewer #2: I Don't Know

Reviewer #3: (No Response)

Reviewer #4: Yes

4. Have the authors made all data underlying the findings in their manuscript fully available?

Reviewer #1: No

Reviewer #2: Yes

Reviewer #3: Yes

Reviewer #4: Yes

5. Is the manuscript presented in an intelligible fashion and written in standard English?

Reviewer #1: No

Reviewer #2: Yes

Reviewer #3: Yes

Reviewer #4: Yes

6. Review Comments to the Author

Reviewer #1: I am not sure the identified predictors for COVID-19 can be precisely predict the Vulnerability to COVID-19. The authors may explore or discuss using machine learning/deep learning to see their prediction performance. For this reason, the following machine-learning based prediction model can be mimicked and the paper can be cited:

Ref 1: Liu, M. et al. A multi-model deep convolutional neural network for automatic hippocampus segmentation and classification in Alzheimer's disease. NeuroImage 208, 116459, doi:10.1016/j.neuroimage.2019.116459 (2020).

Ref 2:Yu, H. et al. LEPR hypomethylation is significantly associated with gastric cancer in males. Experimental and molecular pathology 116, 104493, doi:10.1016/j.yexmp.2020.104493 (2020).

The causal effects of Non-communicable diseases (NCDs) on Vulnerability to COVID-19 should be explored or discussed with mendelian randomization analysis. For this reason,the following papers can be cited or mimicked for the analysis or discussion:

Ref 1:Zhang, F. et al. Causal influences of neuroticism on mental health and cardiovascular disease. Human genetics, doi:10.1007/s00439-021-02288-x (2021).

Ref 2:Zhang, F. et al. Genetic evidence suggests posttraumatic stress disorder as a subtype of major depressive disorder. The Journal of clinical investigation, doi:10.1172/jci145942 (2021).

Ref 3:Wang, X. et al. Genetic support of a causal relationship between iron status and type 2 diabetes: a Mendelian randomization study. The Journal of clinical endocrinology and metabolism, doi:10.1210/clinem/dgab454 (2021).

Ref 4:Hou L, Xu M, Yu Y, Sun X, Liu X, Liu L, Li Y, Yuan T, Li W, Li H, Xue F. Exploring the causal pathway from ischemic stroke to atrial fibrillation: a network Mendelian randomization study.Mol Med. 2020 Jan 15;26(1):7. doi: 10.1186/s10020-019-0133-y.

Reviewer #2: (No Response)

Reviewer #3: Authors have addressed some previous concerns with the manuscript. Overall, much improved submission with a focused attention to hypertension and diabetes mellitus as it relates to vulnerable populations in these regions of Ethiopia during the pandemic.

Reviewer #4: This revised version submitted by Bojola et al has included the reviewer recommended correction in the methods sections, that is defining the poverty line, and in the discussion section, to discuss the biological mechanisms associated with diseases severity of COVID-19 among diabetes and hypertension conditions. It shows much improvement since the original submission. From the public health point of view, it is important to know, how knowledge and preventive practice are important to living a disease-free life.

7. PLOS authors have the option to publish the peer review history of their article (what does this mean?). If published, this will include your full peer review and any attached files.

Reviewer #1: No

Reviewer #2: **Yes: **Yuuki bamba

Reviewer #3: **Yes: **Norman Beatty, MD, University of Florida College of Medicine, Gainesville, FL, USA

Reviewer #4: No

---

## [Author Response · Author response to Decision Letter 2]

24 Oct 2021

Dear reviewer, thank you for providing the references. We have read them and found they are important. However, most of them are experimental study designs to identify the genetic causal link between an outcome variable and factors. However, our study is not about the causal link between diabetes or hypertension and COVID-19. The overall objective of this study was to assess the knowledge and preventive practices of the COVID-9 among patients with hypertension or diabetes. We used a community-based cross-sectional study design with a multi-stage random sampling technique and selected sufficient and representative sample from the three large zones of the Southern region of Ethiopia. Using logistic regression analysis, we have identified several independent predictors of knowledge and preventive practices of COVID-19. The magnitude of the association between the outcome and each independent variable was identified. Based on the study findings, we have forwarded recommendations to adopt major preventive practices to prevent the contraction of the virus. Furthermore, we suggested the Ministry of Health of Ethiopia to give priority to COVID-19 vaccination to these patients with underlying disease conditions. 

Thank you so much.

Aleme Mekuria (MPH/RH, PhD. Fellow)

---

## [Decision Letter · Decision Letter 3]

3 Jan 2022

Non-communicable diseases (NCDs) and Vulnerability to COVID-19: the case of adult patients with Hypertension or Diabetes Mellitus in Gamo, Gofa, and South Omo zones in Southern Ethiopia

PONE-D-20-31282R3

Dear Dr. Mekuria,

We’re pleased to inform you that your manuscript has been judged scientifically suitable for publication and will be formally accepted for publication once it meets all outstanding technical requirements.

Kind regards,

Tauqeer Hussain Mallhi, Ph.D

Academic Editor

PLOS ONE

Additional Editor Comments (optional):

Reviewers' comments:

Reviewer's Responses to Questions

**Comments to the Author**

1. If the authors have adequately addressed your comments raised in a previous round of review and you feel that this manuscript is now acceptable for publication, you may indicate that here to bypass the “Comments to the Author” section, enter your conflict of interest statement in the “Confidential to Editor” section, and submit your "Accept" recommendation.

Reviewer #2: All comments have been addressed

Reviewer #3: All comments have been addressed

Reviewer #4: All comments have been addressed

2. Is the manuscript technically sound, and do the data support the conclusions?

Reviewer #2: Yes

Reviewer #3: Yes

Reviewer #4: Yes

3. Has the statistical analysis been performed appropriately and rigorously? 

Reviewer #2: I Don't Know

Reviewer #3: I Don't Know

Reviewer #4: Yes

4. Have the authors made all data underlying the findings in their manuscript fully available?

Reviewer #2: Yes

Reviewer #3: Yes

Reviewer #4: Yes

5. Is the manuscript presented in an intelligible fashion and written in standard English?

Reviewer #2: Yes

Reviewer #3: Yes

Reviewer #4: Yes

6. Review Comments to the Author

Reviewer #2: Authors have adequately addressed my recommendations.

This study suggests that patients with hypertension or diabetes may be a vulnerable group to COVID-19 that should be intensively intervened by the government.

It is great that you have actually made policy recommendations based on the results of this study.

Reviewer #3: Author's have addressed previous recommendations from the reviewers. Data collected reflects the study aims and objective. Thank you for you work and attempt to break barriers for those in southern Ethiopia impacted by COVID-19.

Reviewer #4: Submitted re-revised manuscript by Bojola et al, to address the importance of knowledge about COVID-19 and preventive practices among subjects with hypertension and diabetes in regions of Southern Ethiopia, has included the reviewer recommended suggestions and modified the manuscript appropriately. Overall, this manuscript clearly shows the relationship between COVID-19 related knowledge and preventive practices, as well as the factors associated to adopt preventive practices among subjects with non-communicable diseases. Such studies are essential to prevent the disease from spreading among vulnerable subject groups. This revised manuscript shows better improvement over the earlier submission and can be accepted for publication. However, there are a few minor recommendations, such as:

1. In Line 306-307, the numbers are redundant.

2. In Line 338, the AOR mentioned in the text is not matching with table

3. In Table 2 and Table 3, observed numbers and percentages are to be distinguished.

7. PLOS authors have the option to publish the peer review history of their article (what does this mean?). If published, this will include your full peer review and any attached files.

Reviewer #2: **Yes: **Yuuki Bamba, MD, PhD

Reviewer #3: No

Reviewer #4: No

---

## [Editor Report · Acceptance letter]

10 Jan 2022

PONE-D-20-31282R3 

Non-communicable diseases (NCDs) and Vulnerability to COVID-19: the case of adult patients with Hypertension or Diabetes Mellitus in Gamo, Gofa, and South Omo zones in Southern Ethiopia 

Dear Dr. Mekuria:

I'm pleased to inform you that your manuscript has been deemed suitable for publication in PLOS ONE. Congratulations! Your manuscript is now with our production department. 

Kind regards, 

on behalf of

Dr. Tauqeer Hussain Mallhi 

Academic Editor

PLOS ONE